# A propensity score-matched analysis of mortality in solid organ transplant patients with COVID-19 compared to non-solid organ transplant patients

Laura Linares[1], Frederic Cofan[2], Fritz Diekmann[2], Sabina Herrera[1], María Angeles Marcos[3], María Angeles Castel[4], Marta Farrero[4], Jordi Colmenero[5], Pablo Ruiz[5], Gonzalo Crespo[5], Jaume Llopis[6], Carolina Garcia-Vidal[1], Àlex Soriano[1], Asunción Moreno[1], Marta Bodro[1] *, on behalf of Hospital Clínic COVID-19 research group[¶]

1 Department of Infectious Diseases, Hospital Clinic – IDIBAPS, ISGlobal (Institute for Global Health), University of Barcelona, Barcelona, Spain, 2 Department of Nephrology and Renal Transplantation, Hospital Clinic – IDIBAPS, ISGlobal (Institute for Global Health), University of Barcelona, Barcelona, Spain, 3 Department of Microbiology, Hospital Clinic – IDIBAPS, ISGlobal (Institute for Global Health), University of Barcelona, Barcelona, Spain, 4 Heart Failure and Heart Transplant Unit, Hospital Clinic – IDIBAPS, ISGlobal (Institute for Global Health), University of Barcelona, Barcelona, Spain, 5 Liver Transplant Unit, Hospital Clinic – IDIBAPS, ISGlobal (Institute for Global Health), University of Barcelona, Barcelona, Spain, 6 Genetic, Microbiology and Statistics Department, Hospital Clinic – IDIBAPS, ISGlobal (Institute for Global Health), University of Barcelona, Barcelona, Spain

¶ Members of the Hospital Clínic COVID-19 research group are provided in the Acknowledgments section
* mbodro@clinic.cat

**Data Availability Statement:** All relevant data are within the manuscript and its Supporting information files.

## Abstract

In the context of COVID-19 pandemic, we aimed to analyze the epidemiology, clinical characteristics, risk factors for mortality and impact of COVID-19 on outcomes of solid organ transplant (SOT) recipients compared to a cohort of non transplant patients, evaluating if transplantation could be considered a risk factor for mortality. From March to May 2020, 261 hospitalized patients with COVID-19 pneumonia were evaluated, including 41 SOT recipients. Of these, thirty-two were kidney recipients, 4 liver, 3 heart and 2 combined kidney-liver transplants. Median time from transplantation to COVID-19 diagnosis was 6 years. Thirteen SOT recipients (32%) required Intensive Care Unit (ICU) admission and 5 patients died (12%). Using a propensity score match analysis, we found no significant differences between SOT recipients and non-transplant patients. Older age (OR 1.142; 95% [CI 1.08–1.197]) higher levels of C-reactive protein (OR 3.068; 95% [CI 1.22–7.71]) and levels of serum creatinine on admission (OR 3.048 95% [CI 1.22–7.57]) were associated with higher mortality. The clinical outcomes of SARS-CoV-2 infection in our cohort of SOT recipients appear to be similar to that observed in the non-transplant population. Older age, higher levels of C-reactive protein and serum creatinine were associated with higher mortality, whereas SOT was not associated with worse outcomes.

**Funding:** The authors received no specific funding for this work.

**Competing interests:** The authors have declared that no competing interests exist.

## Introduction

Severe Acute Respiratory Syndrome Coronavirus 2 (SARS-CoV-2) emerged in December 2019 in China rapidly evolving to the Coronavirus Disease 2019 (COVID-19) pandemic [1]. Reported associated mortality is around 7%, mostly related to older age, obesity, hypertension and chronic pulmonary disease [2].

Emerging data of the impact of COVID-19 in immunosupressed patients, including solid organ transplant (SOT recipients) has recently become available [3–18]. SOT recipients may be at a greater risk for worse outcomes due to the detrimental effect of the immunosuppressive therapy, similar to other viral infections, however, while some studies reflected poorer outcomes [6,9,16–18], other studies do not suggest worse prognosis compared to the non-transplant population [3–5,7,8,15,19].

Furthermore, besides immunosuppressive regimens, transplant recipients usually present more co-morbidities such as hypertension and diabetes, possibly influencing the outcome of patients with COVID-19. On the other hand, a propensity score match analysis has not been performed to assess mortality of solid organ recipients compared to non-solid organ transplant patients before.

In this setting, we aimed to study the epidemiology, clinical characteristics and mortality risk factors of SOT recipients that required hospitalization due to COVID-19 compared to a cohort of non transplant patients hospitalized in the same period, using a propensity score match analysis.

## Material and methods

### Patient selection

From 6th of March to 24h of May, data regarding epidemiology, clinical and laboratory findings and outcomes of patients admitted with respiratory symptoms and radiological evidence of SARS-CoV-2 pneumonia to Hospital Clinic of Barcelona was prospectively recorded. We included all hospitalized patients.

### Clinical data and definitions

Laboratory diagnosis of SARS-CoV-2 infection was made by a positive reverse transcriptase-polymerase chain reaction (RT-PCR) assay from a nasopharyngeal swab.

We used the World Health Organization clinical Ordinary Scale Determination (OSD) to assess patient clinical status. This OSD was recorded at baseline and every day while hospitalization. The ordinal scale categories are as follows: 0) Uninfected 1) Ambulatory patients with no limitation of activities 2) Ambulatory patients with limitation of activities, 3) Patients requiring hospitalization in a non-ICU ward not requiring supplemental oxygen, 4) Patients requiring hospitalization in a non-ICU ward requiring supplemental oxygen, 5) Patients hospitalized in ICU or non-ICU hospital ward, requiring non-invasive ventilation or high-flow oxygen, 6) Patients hospitalized in ICU requiring intubation and mechanical ventilation, 7) Patients hospitalized in ICU, requiring Extracorporeal membrane oxygenation (ECMO) or mechanical ventilation and additional organ support (e.g. vasopressors, renal replacement therapy) and 8) Death during hospitalization.

The following laboratory measurements were recorded: Total lymphocyte count (cell/mm), serum C-reactive protein (mg/dL) (NR<0, D- dimer (ng/mL) (NR<500), serum Lactate dehydrogenase (IU/mL) (NR<234), serum Creatinine, (mg/dL) (NR 0.3–1.3), serum Ferritin (ng /mL) (NR 20–400), serum Troponin (ng /mL) (NR <42.2). For the purpose of statistical

analysis, the highest value recorded during hospitalization was used. For the total lymphocyte count the lowest laboratory value was selected for analysis purposes.

*Streptococcus pneumoniae* co-infection was considered if cultures from respiratory tract samples or urinary antigen were positive together with a chest X-ray or CT-scan suggestive of bacterial pneumonia.

All patients are prescribed prophylactic anticoagulation with enoxaparin 40 mg/24h unless contraindicated. Patients with overweight (>80Kg), D-dimer >3000 ng/mL, or with additional risk factors (cancer, history of thrombosis, recent surgery, etc) are prescribed enoxaparin 60 mg/24h unless contraindicated. In patients with clinical suspicion of pulmonary embolism a chest-CT is performed.

Cryptogenic Organizing pneumonia (COP) was defined on chest CT by multifocal ground glass opacities and/or consolidation.

Biopsy proven acute rejection episodes were recorded in the 3 months prior to admission.

All patients were followed-up after discharge for 2 months.

## Treatment protocol

Our hospital protocol consisted of lopinavir/ritonavir 400/100 mg BID for 7–14 days plus hydroxychloroquine 400 mg/12h on the first day, followed by 200 mg/12h for the next 4 days. Patients with major drug interactions did not receive lopinavir/ritonavir. From the 18th of March, azithromycin 500 mg the first day and 250 mg/24h for 4 additional days was added to the regimen. All patients received prophylactic doses of heparin. The local indication of anti-cytokine therapy was for patients with pneumonia, progressive respiratory failure (increasing fraction of inspired Oxygen) and C-reactive protein (CRP) $\geq$ 8 mg/dL or ferritin $\geq$800 ng/mL or lymphocyte count < 800 cells/mm$^3$.

Choice of anti-cytokine therapy was at the discretion of the attending physician. Available anti-cytokine therapy in our center included: tocilizumab, anakinra and barticinib. Remdesivir therapy was prescribed if patients presented with: a) 7 days of symptoms or less, b)room air oxygen saturation of < 94%, c) glomerular filtration >30 (mL/min), and d) liver function tests < 5 times the upper normal limit according to Spanish Protocol from AEMS (Spanish Agency of Drugs and Heath Products) in addition to the standard of care. Hepatitis B serologies (hepatitis B surface antigen) and QuantiFERON-TB® were performed prior to anti-cytokine prescription and prophylaxis with entecavir and isoniazid were prescribed if applicable.

## Immunosuppressive protocol

At baseline transplant patients could either be in a: a) calcineurin inhibitors based regimen (including tacrolimus or cyclosporine plus a cycle cell inhibitor and prednisone) or b) mTOR based regimen (including everolimus or sirolimus plus cycle cell inhibitor and prednisone).

According to center policy, due to the potential severity of SARS-CoV-2 infection, mycophenolate and mTOR inhibitor (mTORi) (Sirolimus or everolimus) were initially withdrawn in all admitted SOT recipients with COVID-19. Furthermore, in those patients starting treatment with lopinavir/ritonavir, the calcineurin inhibitor (CNI) (tacrolimus or cyclosporine) was also temporary discontinued due to the strong significant increase of CNI levels. Maintenance immunosuppression consisted of prednisone monotherapy (10–20 mg/day) until COVID-19 resolution, at which time tacrolimus was reinitiated at reduced doses (through blood levels around 5 ng/mL).

## Statistical analysis

In the comparative analysis, we used the chi-square test with Yate's correction for categorical variables. Depending on their homogeneity, continuous variables were compared using the *t* test or Mann-Whitney test. Statistically significant variables in the univariate analysis including median age and sex were entered into a multivariate model using logistic regression analysis, and the odds ratios (OR) and 95% confidence intervals (CI) were calculated.

Propensity score matching was calculated using the following parameters: age, sex, hypertension, lung disease, use of anti-cytokine therapies, baseline OSD and OSD during hospitalization.

The analysis was performed using the stepwise logistic regression model of the SPSS software package (SPSS version 23.0, SPSS Inc., Chicago, Illinois, USA). All statistical tests were 2-tailed, and the threshold of statistical significance was set at $p < 0.05$.

The Institutional Ethics Committee of the Hospital Clínic of Barcelona, approved the study and due to the nature of retrospective chart review, waived the need for inform consent from individual patients (Comité Ètic d'Investigació Clínica; HCB/2020/0273). All patients signed an informed consent for therapies off-label use.

## Results

Two hundred and sixty-one patients were included in our study. Forty-one of them were SOT recipients, including 32 kidney recipients (78%), 4 liver recipients (10%), 3 heart recipients (7%) and 2 combined liver-kidney recipients (5%). Median follow-up was 68 days (IQR 57–75). Table 1 shows SOT recipients baseline characteristics. Median years from transplantation to COVID-19 diagnosis were 6 (range, 1–21). Fever was found in 95% (39) of the patients followed by cough in 68% (28) and dyspnoea in 32% (13). Two patients presented *Streptococcus pneumoniae* co-infection.

Table 2 shows the clinical characteristics and laboratory findings of transplant and non-transplant patients with COVID-19. We compared baseline characteristics of both groups. Hypertension was more frequent in transplanted patients compared to the non-transplant group (81% vs 45%, p<0.001), whereas diabetes mellitus was more frequent in the non-transplant group (32% vs 16%, p = 0.01). Transplant recipients had significantly more chronic kidney disease than non-transplant patients (34% vs 5% p <0.001). Dyspnoea and cough on admission were more frequent in the non-transplant group (57% vs 32%, p = 0.003; 82% vs 68%, p = 0.032). Median ferritin levels on admission were higher in the non-transplant group than in SOT recipients (776 vs 321, p = 0.03). Transplant patients had higher serum creatinine levels on admission than the non-transplant group (p<0.001). Regarding COVID-19 specific treatment, 23 transplant recipients received at least one anti-cytokine therapy (57%) that included tocilizumab, anakinra or baricitinib, compared to 125 (57%) of the non-transplant group (p = 0.932).

We found several differences in the complications during hospitalization; acute kidney injury was more frequent in the transplant group (49% vs. 16%, p<0.001). Persistent lymphopenia was more frequent in the transplant cohort compared to the non-transplant group (p = 0.001). Four cases of cryptogenetic organizing pneumonia were registered in the SOT group (10%) compared to 42 (19%) in the non-transplant group (p = 0.1). None of the SOT recipients was diagnosed with pulmonary embolism whereas it was diagnosed in 21 patients of the non-transplant group (0% vs 10%, p = 0.04). Transplant patients had longer hospital stay compared to the non-transplant group, almost reaching statistical significance (17 days vs 12, p = 0.054). There were no differences in terms of mortality between both groups (15% vs 12%, p = 0.64). Respiratory failure due to COVID-19 was the main cause of death in all patients

**Table 1. Baseline characteristics of SOT recipients with COVID-19.**

| Variables | n = 41 (%) |
|---|---|
| Median age, years, IQR | 58 (33–86) |
| Male sex | 27 (66) |
| Transplanted organ | |
| Kidney | 32 (78) |
| Liver | 4 (10) |
| Heart | 3 (7) |
| Combined liver-kidney transplant | 2 (5) |
| Underlying conditions | |
| Hypertension | 33 (81) |
| Diabetes mellitus | 34 (83) |
| Cardiovascular disease | 10 (24) |
| COPD | 8 (20) |
| Chronic kidney disease | 14 (34) |
| Years from transplant to diagnosis, median (IQR) | 6 (1–21) |
| Immunosupressive regimen | |
| Calcineurin inhibitors based therapy | 26 (63) |
| mTOR based therapy | 15 (37) |
| Previous episodes of acute rejection (3 months, only biopsy proven) | 0 |
| COVID-19 adjuvant treatment | |
| Lopinavir/ritonavir | 31 (76) |
| Hydroxychloroquine | 40 (98) |
| Remdesivir | 0 |
| Azithromycin | 41 (100) |
| Tocilizumab | 19 (46) |
| Anakinra | 7 (17) |
| Baricitimib | 1 (2) |
| Steroids pulse | 17 (41) |
| Interferon | 3 (7) |

*COPD: Chronic obstructive pulmonary disease.

with the exception of a patient that died of complications of infective endocarditis. Fig 1 shows the Kaplan-Meier curve.

Table 3 shows clinical Ordinary Scale Determination on admission and during hospitalization by transplantation.

In terms of outcomes, 14 patients in the transplant group required ICU admission, of them 7 patients required mechanical ventilation (50%) and 5 died (12%). Ninety patients of the non-transplant group required ICU admission. Of them, 43 (20%) required mechanical ventilation and 33 died (15%).

Multivariate analysis of risk factors for mortality was performed and is depicted in Table 4. Older age (OR 1.142; 95% [CI 1.08–1.197]) higher levels of serum C-reactive protein (OR 3.068; 95% [CI 1.22–7.71]) and higher levels of serum creatinine on admission (OR 3.048 95% [CI 1.22–7.57]) were associated with mortality.

We performed a propensity score matching (PSM) including 36 patients in each group (Table 5).

Patients were matched one-to-one using PSM to eliminate confounding factors. Clinical variables entered into the PSM analysis were age, gender, comorbidities, biological therapy,

**Table 2. Clinical characteristics and laboratory findings of transplant and non-transplant patients with COVID-19.**

| Variables | Transplanted n = 41 (%) | Non-transplanted n = 220 (%) | p |
|---|---|---|---|
| Age in years, median (IQR) | 58 (33–86) | 63 (51–72) | 0.175 |
| Male sex | 27 (66) | 144 (66) | 0.961 |
| Underlying conditions | | | |
| Hypertension | 33 (81) | 98 (45) | <0.001 |
| Diabetes mellitus | 34 (16) | 13 (32) | 0.013 |
| Cardiovascular disease | 10 (24) | 29 (13) | 0.065 |
| COPD | 8 (20) | 38 (17) | 0.730 |
| Chronic kidney disease | 14 (34) | 11 (5) | < 0.001 |
| Symptoms at admission | | | |
| Fever | 39 (95) | 201 (91) | 0.417 |
| Diarrhea | 9 (22) | 48 (22) | 0.985 |
| Dyspnoea | 13 (32) | 126 (57) | 0.003 |
| Cough | 27 (68) | 181 (82) | 0.032 |
| Median days from symptoms to diagnosis (IQR) | 5 (2–9) | 6 (4–8) | 0.276 |
| Laboratory values on admission, median (IQR) | | | |
| Lymphocyte (cell/mm) | 600 (400–900) | 800 (600–1000) | 0.220 |
| C-reactive protein (mg/dL) (NR<0) | 9.4 (3–13.8) | 8.1 (4.2–16.8) | 0.479 |
| D- dimer (ng/mL) (NR<500) | 800 (600–2447) | 800 (450–1400) | 0.706 |
| Lactate dehydrogenase (IU/mL) (NR<234) | 289(400–900) | 338 (264–424) | 0.390 |
| Creatinine, (mg/dL) (NR 0.3–1.3) | 1.8 (1.2–2.9) | 0.9 (0.7–1.09) | <0.001 |
| Ferritin (ng /mL) (NR 20–400) | 321 (264–949) | 776 (391–1421) | 0.030 |
| Troponin (ng /mL) (NR <42.2) | 8.2 (4.7–30.7) | 11.2 (5.1–20.5) | 0.551 |
| COVID-19 adjuvant treatment | | | |
| Lopinavir/ritonavir | 31 (76) | 205 (93) | 0.001 |
| Hydroxychloroquine | 40 (98) | 216 (98) | 0.390 |
| Azithromycin | 41 (100) | 220 (100) | 0.932 |
| Tocilizumab | 19 (46) | 125 (57) | 0.216 |
| Anakinra | 7 (17) | 4 (2) | <0.001 |
| Baricitinib | 1 (2) | 0 | 0.020 |
| Remdesivir | 0 | 29 (13) | 0.005 |
| Anti-cytokine therapy | 23 (56) | 125 (57) | 0.932 |
| Clinical outcomes | | | |
| Intensive care unit admission | 14 (34) | 90 (41) | 0.317 |
| • non-invasive ventilation | 7 (17) | 47 (21) | 0.928 |
| • mechanical ventilation | 7 (17) | 43 (19) | 0.325 |
| Complications | | | |
| Acute kidney injury | 20 (49) | 35 (16) | 0.001 |
| Other infection | 10 (24) | 26 (19) | 0.462 |
| Septic shock | 5 (12) | 51 (23) | 0.110 |
| Organizing pneumonia | 4 (10) | 42 (19) | 0.150 |
| Pulmonary embolism | 0 | 21 (10) | 0.04 |
| Higher/lower* laboratory values during admission, median (IQR) | | | |
| Lymphocyte (cell/mm) | 700 (500–1100) | 1600 (1200–2300) | 0.001 |
| C-reactive protein (mg/dL) | 14.4 (9–18.9) | 15.5 (8.8–22.6) | 0.754 |
| D-dimer (ng/mL) | 1300 (1000–4010) | 2400 (1000–7600) | 0.190 |
| Lactate dehydrogenase (IU/mL) | 367 (285–525) | 448 (348–567) | 0.027 |
| Ferritin (ng /mL) | 888 (442–1515) | 1135 (648–1934) | 0.161 |

(*Continued*)

**Table 2.** (Continued)

| Variables | Transplanted n = 41 (%) | Non-transplanted n = 220 (%) | p |
|---|---|---|---|
| Troponin (ng/mL) | 15.2 (7.1–52.0) | 13.7 (6.3–31.2) | 0.860 |
| Median days of hospitalization (IQR) | 17 (12–24) | 12 (8–20) | 0.054 |
| Median days in ICU, median (IQR) | 11 (7–22) | 9 (5–15) | 0.303 |
| Mortality | 5 (12.2) | 33 (15) | 0.640 |

* In case of C-reactive protein, D-dimer, lactate dehydrogenase, ferritin and troponin we selected the highest laboratory value during hospitalization. And in case of lymphocyte count we select the lowest laboratory value.

categorical ordinal scale and mortality. There were no significant differences between matched groups in terms of mortality (14% vs 17%).

## Discussion

In our study comparing a cohort of hospitalized SOT recipients with cohort of non-transplant recipients using a propensity score analysis we found no differences in mortality between the two groups. Therefore, in our cohort of patients with COVID-19, SOT cannot be considered a risk factor for mortality.

In our cohort of COVID-19 hospitalized patients, we found no differences in clinical presentation between SOT and non-transplant patients. However, the true incidence of COVID-19 infection and symptom development in SOT vs. general population remains to be established, due to the lack of universal testing in asymptomatic recipients. Ongoing seroprevalence studies may provide information on true infection incidence.

We did find differences in the use of COVID-19 specific therapy that was considered standard of care at the time of the study in the two groups. Transplant patients received significantly less lopinavir/ritonavir, and more anakinra compared to non-transplant patients. The important drug-drug interactions of lopinavir/ritonavir with immunosuppressant medications could have influenced physicians prescribing this drug [20]. Remdesivir which has been

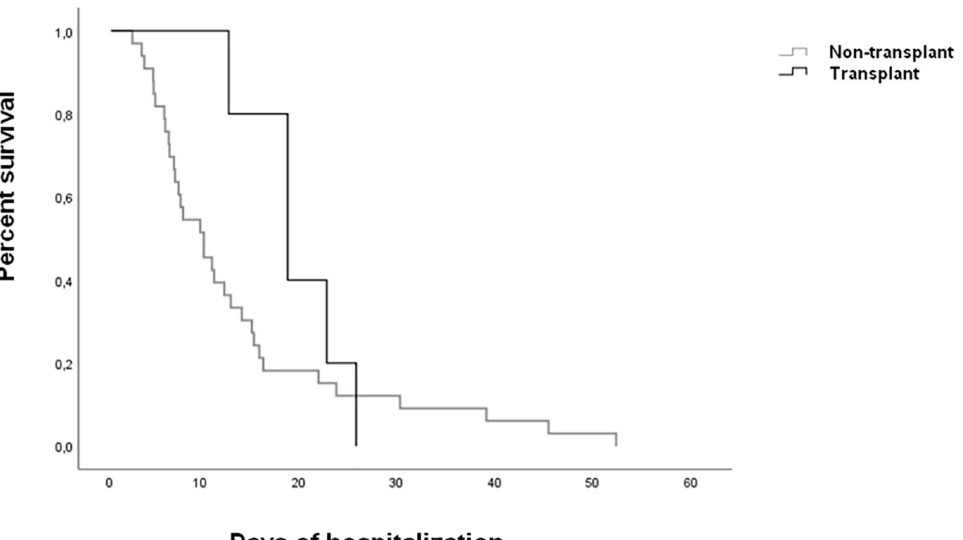

**Fig 1. Kaplan-Meier analysis by transplant.**

**Table 3. Ordinal Scale Determination at baseline, during hospitalization and at end of follow up by transplant.**

| Ordinal Scale Determination | Transplant n = 41 (%) | Non-transplant n = 220 (%) |
|---|---|---|
| **Baseline** | | |
| Non-ICU[a] hospital ward, not requiring supplemental oxygen | 25 (61) | 63 (29) |
| Non-ICU hospital ward requiring supplemental oxygen | 13 (32) | 135 (61) |
| ICU or non-ICU hospital ward requiring non-invasive ventilation or high-flow oxygen | 2 (5) | 1 (1) |
| ICU, requiring intubation and mechanical ventilation | 1 (2) | 21 (10) |
| **During hospitalization** | | |
| Non-ICU hospital ward, not requiring supplemental oxygen | 9 (22) | - |
| Non-ICU hospital ward requiring supplemental oxygen | 18 (44) | 130 (59) |
| ICU or non-ICU hospital ward requiring non-invasive ventilation or high-flow oxygen | 7 (17) | 47 (21) |
| ICU, requiring intubation and mechanical ventilation or ECMO[b] | 7 (17) | 43 (20) |
| **End of follow-up** | | |
| Discharged from hospital | 36 (88) | 189 (86) |
| Death | 5 (12) | 33 (15) |

[a]ICU, intensive care unit;

[b]ECMO, *Extracorporeal membrane oxygenation*.

shown to reduce the time to clinical improvement, was not administered to SOT patients, possibly as a result of higher incidence of chronic kidney disease and acute kidney injury, that contraindicate the use of this drug [21]. Randomized control trials with lopinavir/ritonavir and azythromycin as adyuvant therapy for COVID-19 have shown no benefit, and some have shown deleterious effects, and therefore they are no longer being used [22].

To the date, remdesivir and dexamethasone have shown to be beneficial in randomized controlled trials in the treatment of patients with COVID-19 [23–27]. Preliminary studies found promising results with the use of anakinra [28–32] and other anti-cytokine therapies [33–35], however, randomized controlled trials analyzing the use of these therapies the SOT population are lacking, with the special interest of development of opportunistic infections. In terms of complications during hospitalization, we found that acute kidney failure was more common in the transplant cohort, probably reflecting kidney function vulnerability of SOT recipients, and use of calcineurin inhibitors. Transplant patients had lymphopenia for a significantly longer time compared to non-transplant patients, possibly as a consequence of the use of drugs that cause myelotoxicity such as maintenance immunosuppressive therapy and prophylactic antibiotics. It is to be noted that despite having 25% patients with septic shock in the SOT group, compared to 12% in the non-transplant group, this did not influence the mortality significantly.

In the multivariate analysis of risk factors of mortality, solid organ transplantation was not associated with death. Our mortality rate is similar to other transplant cohorts [3,4,15]. However, other cohort have reported much worse outcomes, with mortality ranging from 28 to 67% [10,12]. These differences might me explained by demographic differences, including limited heath resources in the context of an overstretched health system in the peak of the pandemic. There have been several discussions around the impact of immunosupression on COVID-19 prognosis. It is reasonable to assume that during the viremic phase immunosuppressive therapy could potentially be deleterious, however, some studies have found that some drugs such as cyclosporine, tacrolimus and mTOR inhibitors [36] have in vitro activity against

**Table 4. Univariate and multivariate analysis of risk factors associated with mortality.**

| | Category | n | Mortality n (%) | Univariate analysis | | Multivariate analysis | |
|---|---|---|---|---|---|---|---|
| | | | | OR (95% CI) | p value | OR (95% CI) | p value |
| Gender | Male | 171 | 23 (13.5) | | | | |
| | Female | 90 | 15 (16.7) | 1.287 (0.634–2.611) | 0.484 | | |
| Age, years | < 63 | 137 | 4 (2.9) | | | | |
| | > 63 | 124 | 34 (27.4) | 12.561 (4.309–36.62) | <0.001 | 1.142 (1.089–1.197) | <0.001 |
| Hypertension | Yes | 131 | 27 (20.6) | | | | |
| | No | 130 | 11 (8.5) | 2.809 (1.328–5.939) | 0.005 | | |
| Cardiovascular disease | Yes | 39 | 14 (35.9) | | | | |
| | No | 222 | 24 (10.8) | 4.620 (2.119–10.073) | <0.001 | | |
| Chronic respiratory disease | Yes | 46 | 13 (28.3) | | | | |
| | No | 215 | 25 (11.6) | 2.994 (1.393–6.436) | 0.004 | | |
| Solid Organ Transplantation | Yes | 41 | 5 (12.2) | | | | |
| | No | 220 | 33 (15) | 0.787 (0.288–2.152) | 0.640 | | |
| serum C-reactive protein (mg/dL) Day 0 | < 8.2 | 130 | 15 (11.5) | | | | |
| | >8.2 | 131 | 23 (17.6) | 1.633 (0.810–3.293) | 0.168 | | |
| serum C-reactive protein (mg/dL), maximum value | < 15.2 | 130 | 8 (6.2) | | | | |
| | >15.2 | 131 | 30 (23) | 4.530 (1.989–10.318) | <0.001 | 3.068 (1.22–7.71) | 0.017 |
| serum Creatinine (mg/dL) Day 0 | < 0.9 | 121 | 10 (8.3) | | | | |
| | > 0.9 | 140 | 28 (20) | 2.775 (1.287–5.983) | 0.007 | 3.048 (1.226–7.575) | 0.016 |
| Lymphocytes (cell/mm3) Day 0 | < 700 | 133 | 24 (18) | | | | |
| | >700 | 128 | 14 (36.8) | 0.558 (0.274–1.134) | 0.104 | | |
| Serum ferritin (ng/mL) Day 0 | < 749 | 124 | 14 (11.3) | | | | |
| | >749 | 137 | 24 (17.5) | 1.669 (0.821–3.393) | 0.154 | | |
| D-dimer (ng/mL) Day 0 | < 800 | 140 | 13 (9.3) | | | | |
| | > 800 | 121 | 25 (20.7) | 2.544 (1.237–5.230) | 0.009 | | |
| Intensive care unit (ICU) admission | Yes | 101 | 21 (20.8) | | | | |
| | No | 160 | 17 (10.6) | 2.208 (1.101–4.427) | 0.023 | | |
| Invasive mechanical ventilation | Yes | 53 | 12 (22.6) | | | | |
| | No | 208 | 26 (12.5) | 2.049 (0.955–4.395) | 0.062 | | |
| Septic shock | Yes | 56 | 14 (25) | | | | |
| | No | 203 | 23 (11.3) | 2.609 (1.239–5.492) | 0.010 | | |
| Biological therapy | Yes | 148 | 18 (47.4) | | | | |
| | No | 113 | 130 (58.3) | 0.644 (0.323–1.284) | 0.209 | | |
| OSD[a], baseline | | | | | | | |
| Conventional ward, not requiring supplemental oxygen | Yes | 88 | 5 (5.7) | | | | |
| Conventional ward, requiring supplemental low-flow oxygen | Yes | 148 | 25 (16.9) | | | | |
| ICU[b], requiring supplemental high-flow supplemental oxygen | Yes | 3 | 2 (66.7) | | | | |
| ICU, requiring invasive mechanical ventilation/ECMO[c] | Yes | 22 | 6 (27.3) | 15.629 | 0.001 | | |

[a]OSD, ordinary scale determination;

[b]ICU, intensive care unit.

[c]ECMO, *Extracorporeal membrane oxygenation.*

other coronaviruses. Furthermore, immunosuppressive regimens might be beneficial preventing or in the event of a cytokine storm. Our cohorts median time from transplant was 6 years, with no documented recent episodes of rejection; and therefore not in the maximum period of immunosuppression.

**Table 5. Comparison of transplant and non-transplant population after applying a propensity score matching.**

| Variables | Transplant n = 36 (%) | Non-transplant n = 36 (%) | p-value |
|---|---|---|---|
| Age in years, media (SD) | 59.6 (13.2) | 60.5 (12.7) | 0.750 |
| Male sex | 11 (31) | 13 (36%) | 0.482 |
| Underlying conditions | | | |
| Hypertension | 28 (78) | 29 (81) | 0.693 |
| Lung disease | 8 (22) | 7 (19) | 0.693 |
| Anti-cytokine therapy | 18 (50) | 20 (56) | 0.514 |
| Mortality | 5 (14) | 6 (17) | 0.640 |

Older age, maximum serum C-reactive protein and serum creatinine levels were associated with mortality. Older age has been related to worse prognosis in several studies [2,37–39]. C-reactive protein, a protein whose expression is driven by IL-6, is a biomarker of severe infection that has been associated with the inflammation cytokine storm related to COVID-19 [40].

Similarly, acute kidney injury has been associated with an increased risk of death in critically ill patients with pneumonia [41,42]. The SOT cohort also had significantly higher rate of chronic kidney disease, without having an impact on the overall mortality.

Our study has several strengths and limitations. We were able to analyze all consecutive admissions during the study period, being one of the first studies comparing transplant patients to non-transplant patients. However, there are several limitations; first of all, as it is a single-centre study, our findings may be attributable to institution-specific variables and secondly, it may not reflect the epidemiology of different centers and/or geographical areas, thus more extensive data are needed to confirm these results. In addition, given the small sample size of our study; we cannot exclude a type 2 error.

To conclude, the clinical course of SARS-CoV-2 infection in SOT recipients appears to be similar to that observed in the non-transplant population, even though COVID-19 specific treatment was different between SOT recipients and non-transplant patients. Older age, serum creatinine and C-reactive protein levels were associated with higher mortality. Solid organ transplant recipients did not experience worse outcomes.

## Supporting information

**S1 File.**
(SAV)

## Acknowledgments

**Hospital Clínic COVID-19 research group**: Albiac L[1], Agüero D[1], Ambrosioni J[1], Blanco JL[1], Cardozo C[1], Chumbita M[1], De la Mora L[1], García-Alcaide F[1], García-Pouton N[1], González-Cordón A[1], Hernández-Meneses M[1], Inciarte A[1], Laguno M[1], Leal L[1], Macaya I[1], Mallolas J[1], Martínez E[1], Martínez M[1], Meira F[1], Miró JM[1], Mensa J[1], Moreno-Martínez A[1], Moreno-García E[1], Morata L[1], Martínez JA[1], Puerta-Alcalde P[1], Rico V[1], Rojas J[1], Solá M[1], Torres B[1], Torres M[1], Ana García[4], Perez-Villa F[4], Navasa M[4], Bayès B[2], Cucchiari D[2], Esforzado N[2], Guillen E[2], Molina A[2], Montagud-Marrahi E[2], Oppenheimer F[2], Piñeiro GJ[2], Poch E[2], Revuelta I[2], Rodas L[2], Torregrosa JV[2], Ugalde-Altamirano J[2], Ventura-Aguiar P[2], Hurtado JC[3], Fernandez M[3], Mosquera MM[3].

## Author Contributions

**Conceptualization:** Laura Linares.

**Data curation:** Laura Linares.

**Formal analysis:** Laura Linares, Jaume Llopis.

**Investigation:** Marta Bodro.

**Methodology:** Jaume Llopis, Marta Bodro.

**Project administration:** Marta Bodro.

**Resources:** Marta Bodro.

**Software:** Marta Bodro.

**Supervision:** Frederic Cofan, Sabina Herrera, Marta Bodro.

**Validation:** Frederic Cofan, Marta Bodro.

**Visualization:** Marta Bodro.

**Writing – original draft:** Laura Linares, Marta Bodro.

**Writing – review & editing:** Fritz Diekmann, Sabina Herrera, María Angeles Marcos, María Angeles Castel, Marta Farrero, Jordi Colmenero, Pablo Ruiz, Gonzalo Crespo, Carolina Garcia-Vidal, Àlex Soriano, Asunción Moreno, Marta Bodro.

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
