## [Decision Letter · Decision Letter 0]

25 Nov 2020

PONE-D-20-31759

No increased risk of mortality in solid organ transplant patients with COVID-19

PLOS ONE

Dear Dr. Bodro,

Thank you for submitting your manuscript to PLOS ONE. After careful consideration, we feel that it has merit but does not fully meet PLOS ONE’s publication criteria as it currently stands. Therefore, we invite you to submit a revised version of the manuscript that addresses the points raised during the review process.

We look forward to receiving your revised manuscript.

Kind regards,

Tzevat Tefik, MD

Academic Editor

PLOS ONE

Journal Requirements:

2. One of the noted authors is a group or consortium [Hospital Clínic COVID-19 research group]. In addition to naming the author group, please list the individual authors and affiliations within this group in the acknowledgments section of your manuscript. Please also indicate clearly a lead author for this group along with a contact email address.

Reviewers' comments:

Reviewer's Responses to Questions

**Comments to the Author**

1. Is the manuscript technically sound, and do the data support the conclusions?

Reviewer #1: Partly

Reviewer #2: Partly

Reviewer #3: Yes

2. Has the statistical analysis been performed appropriately and rigorously? 

Reviewer #1: Yes

Reviewer #2: Yes

Reviewer #3: No

3. Have the authors made all data underlying the findings in their manuscript fully available?

Reviewer #1: Yes

Reviewer #2: Yes

Reviewer #3: Yes

4. Is the manuscript presented in an intelligible fashion and written in standard English?

Reviewer #1: Yes

Reviewer #2: Yes

Reviewer #3: Yes

5. Review Comments to the Author

Reviewer #1: Dear authors,

Thank you for this submission evaluating COVID-19 in solid organ transplantation patients.

I have some major concerns regarding your submission:

I think that the methods section should be changed extensively. Many details given at results section were not defined at methods section. This made me confused when reading results section. I could not clearly understand where some findings came from. Examples of these and my other concerns are given below:

Results section:

-Line 140: Which organs have been transplanted in patients with double transplant? Please define.

- Line 159: What do you mean by biologic therapy? Please define.

-Line 167: p value is higher than 0.05 so it cannot be concluded as higher length of hospital stay in transplant patients.

-Table 1 and Table 2: I am not sure if clinical outcomes section is necessary as you also compared mortality between two groups. The definition of these outcomes is not given. A patient could be transferred to intensive care unit but I cannot understand if s/he was intubated or discharged or died so accepting these as outcomes made me confused.

- What is the difference between median follow-up section in Table 1 and median days of hospitalization? Follow- up is longer than the other variable. Did you follow up patients after discharge? Please define these variables.

- Table 2: What is the criteria for remdesivir therapy? Did some patients receive both remdesivir and lopinavir/ritonavir? If yes, what was the criteria for this condition?

Table 2: What do you mean by higher laboratory values during admission? How did you select parameters in the table? For the parameters, which one did you put in statistical analysis as most probably a patient had multiple lymphocyte, D-dimer, etc. values.

Line 176: Even though emphasized in the abstract, I think less importance was given to Ordinary Scale Determination in the results section. It has been discussed in discussion section. I am not sure if you could compare the groups’ Ordinary Scale Determination statistically but without a “p” value, it is not so valuable.

Table 4: Performing a multi variable comparison is important in studies like this. However, there should be a rationale to create groups and this rationale should be defined at methods section. For example, what was the reason to group age into two according to 63 years? This is valid for C-reactive protein, creatinine, lymphocyte, ferritin, D-dimer.

Table 5: Why did you analyze 36 of transplanted patients? How did you exclude other 5 patients? What was the aim of this test as you have already given all statistical analysis with 41 transplanted patients?

Discussion section:

-Why did not all transplanted patients have lopinavir/ritonavir treatment? Please give some details fort his condition.

-Hydroxychloroquine was advised to be not given at 04 July and lopinavir/ritonavir was advised to be not given at 06 July by World Health Organization. I respect your treatment at that time as most of medical practitioners used these drugs at that time. However, I think that you should add this fact to discussion section

Reviewer #2: My concerns were below.

Introduction

1. Authors should clearly define the purpose of the study and why these authors conducted this research.

2. As I know that solid organ transplant recipients have more comorbid diseases than patients without SOT. The authors should point to this issue in a separate paragraph in this section.

Material and method

1. I recommend authors to use subtitles. Subheadings; patient selection, treatment, outcomes, hospitalization, and ICU admission may use.

2. Inclusion and exclusion criteria should be clearly defined in this section.

3. The selection of the patients without SOT should be defined in this section, and criteria for a propensity score match should be defined.

4. Study endpoints should be clearly stated.

5. Authors should refer to the selection criteria of the patients without SOT.

Results

1. The diabetes rate in Table 1 is misspelled.

2. After the propensity score match, the differences in covid-19 treatment should be explained.

3. Kaplan Meier analysis can be useful for determining differences in mortality between groups.

Patients with macrophage activation syndrome due to COVID-19 can be shown in the tables.

Discussion

1. The mortality rate of SOT recipients with COVID-19 was lower than in some of the previous articles. The authors must find out what factors can be associated with these differences.

2. The authors should discuss the impact of chronic kidney disease on covid 19 mortality in a separate paragraph.

3. Treatment modalities of the patients should be discussed in a separate paragraph.

4. The authors did not perform a power analysis before calculation, and the results may have been determined due to the second type of statistical error analysis. Therefore, authors should not use exact expressions.

Reviewer #3: This manuscript has conclusions similar to other large SOT cohorts that transplant itself does not necessarily increase risk for COVID-19 associated morbidity / mortality, and that other factors such as age and comorbidities play a role. While I agree with a propensity-matched score approach, the study has significant limitations including lack of information on key transplant-related variables (immunosuppression, timing from transplant, rejection) as well as the severity of illness in the non-transplant group.

Line 59: Mortality rate of 7% is higher than most estimates, although there are some studies were case fatality rate was that high. Suggest finding more accurate estimate.

Lines 89-97: Is there a reason that the authors did not use World Health Organization ordinal scale 0-8 that is current reference for clinical severity?

Lines 103: What “anti-cytokine” therapy was used?

Lines 131-132: how were the variables for propensity score matching chosen?

Line 140: What does “double transplant” mean? Is it a patient who has had 2 sequental transplants (e.g. 2 kidney transplants after a single failed grafts) or 2 organs (like kidney-pancreas)?

Line 145: How was it determined that the patients had Strep pneumo co-infection? Was urinary antigen testing or culture used, and what imaging corresponded to make diagnosis?

Line 146: Table 1 should include characteristics for the non-SOT group, with p-values. By definition, outcomes should not be included in Table 1. The authors are also clearly missing some important variables relevant to the degree of immunosuppression of SOT recipients such as time from transplant, induction regimen used, maintenance immunosuppression, presence / absence of rejection, recent treatment for rejection.

Line 172: It is very interesting to compare the characteristics of SOT and non-SOT groups. It appears that the non-SOT group was quite sick with 23% of patients experiencing septic shock, and higher baseline mortality. This should be considered when interpreting the study

Line 260-263: The finding of differences in therapy is not significant other than the fact that it outlines the differences in practice at this one center.

Line 263-265: There ARE therapies that have been shown to be beneficial in RCTs: remdesivir and dexamethasone.

Line 279: The fact that transplant mortality was similar to other centers I think emphasizes that non-transplant mortality in this cohort was high.

Line 298: Authors should include in limitation that there was routine use of agents without proven efficacy, and some with associated harm (protease inhibitors, hydroxychloroquine, anti-cytokine agents) were used as part of the institution’s standard of care.

Comment: This manuscript appears to be written by non-native speaker. Please check for spelling and grammatical errors.

6. PLOS authors have the option to publish the peer review history of their article (what does this mean?). If published, this will include your full peer review and any attached files.

Reviewer #1: No

Reviewer #2: No

Reviewer #3: No

---

## [Author Response · Author response to Decision Letter 0]

23 Dec 2020

Barcelona, December, 2020

Editors

Plos One

Ref.: Ms. No. PONE-D-20-31759

Dear Editors, 

Thank you very much for your review of our manuscript entitled “No increased risk of mortality in solid organ transplant patients with COVID-19”. We truly think that your review has helped us to improve the article. According to your indications we provide a point-by-point response to your comments.

We look forward to hearing from you, 

Sincerely, 

Marta Bodro

Infectious Diseases Department

Hospital Clínic de Barcelona

Barcelona, Spain

martabodro@gmail.com

Review Comments to the Author

Reviewer #1: Dear authors,

Thank you for this submission evaluating COVID-19 in solid organ transplantation patients.

I have some major concerns regarding your submission:

I think that the methods section should be changed extensively. Many details given at results section were not defined at methods section. This made me confused when reading results section. I could not clearly understand where some findings came from. Examples of these and my other concerns are given below:

R: We would like to thank reviewer one for the very useful comments. We have rewritten the methods section, including subheadings, hoping it reads more clearly now. 

Results section:

-Line 140: Which organs have been transplanted in patients with double transplant? Please define.

R: We have included in the results section, that 2 of the SOT patients were combined liver-kidney transplant. Page 7, lines 173

- Line 159: What do you mean by biologic therapy? Please define.

R: We changed the term biologic therapy to “anti-cytokine therapy” throughout the manuscript. Anti-cytokine therapy includes tocicizumab, anakinra and baricitinib, as these were the available drugs in our center. This has been included in the methods section.

-Line 167: p value is higher than 0.05 so it cannot be concluded as higher length of hospital stay in transplant patients.

R: According to your suggestion we clarified that sentence. Page 9, lines 200

-Table 1 and Table 2: I am not sure if clinical outcomes section is necessary as you also compared mortality between two groups. The definition of these outcomes is not given. A patient could be transferred to intensive care unit but I cannot understand if s/he was intubated or discharged or died so accepting these as outcomes made me confused.

R: We removed outcomes from Table 1 and have given exact definitions of the outcomes in the methods section: 

5) Patients hospitalized in ICU or non-ICU hospital ward, requiring non-invasive ventilation or high-flow oxygen, 

6) Patients hospitalized in ICU requiring intubation and mechanical ventilation, 

7) Patients hospitalized in ICU, requiring Extracorporeal membrane oxygenation (ECMO) or mechanical ventilation and additional organ support (e.g. vasopressors, renal replacement therapy)

- What is the difference between median follow-up section in Table 1 and median days of hospitalization? Follow- up is longer than the other variable. Did you follow up patients after discharge? Please define these variables.

R: All patients were followed-up after discharge for 2 months. We included this information at the methods section. Page 5, line 116. 

- Table 2: What is the criteria for remdesivir therapy? Did some patients receive both remdesivir and lopinavir/ritonavir? If yes, what was the criteria for this condition?

R: Remdesivir therapy was prescribed if patients presented with: a) 7 days of symptoms or less, b)room air oxygen saturation of < 94%, c) glomerular filtration >30 (mL/min), and d) liver function tests < 5 times the upper normal limit according to Spanish Protocol from AEMS (Spanish Agency of Drugs and Heath Products) in addition to the standard of care. 

Page 5, lines 129-133. Remdesivir was added to standard of care at the time of the study. We included this information in the Methods section.

Table 2: What do you mean by higher laboratory values during admission? How did you select parameters in the table? For the parameters, which one did you put in statistical analysis as most probably a patient had multiple lymphocyte, D-dimer, etc. values.

R: In case of C-reactive protein, D-dimer, lactate dehydrogenase, ferritin and troponin we selected the highest laboratory value during hospitalization. And in case of lymphocyte count we selected the lowest laboratory value. We clarified this in the method section. Page 4,lines 107-108.

Line 176: Even though emphasized in the abstract, I think less importance was given to Ordinary Scale Determination in the results section. It has been discussed in discussion section. I am not sure if you could compare the groups’ Ordinary Scale Determination statistically but without a “p” value, it is not so valuable.

R: We agree with the reviewer that statistically we could not compare the groups using OSD (we followed the advice of a statistics expert), so, unfortunately we were unable to show a p value. Nevertheless we hope the table illustrates the progress of patients in both groups during hospitalization until discharge. Table 2 includes p values for related outcomes such as need for mechanical ventilation, or death.

Table 4: Performing a multi variable comparison is important in studies like this. However, there should be a rationale to create groups and this rationale should be defined at methods section. For example, what was the reason to group age into two according to 63 years? This is valid for C-reactive protein, creatinine, lymphocyte, ferritin, D-dimer.

R: In the comparative analysis, we used the chi-square test with Yate’s correction for categorical variables. Depending on their homogeneity, continuous variables were compared using the t test or Mann-Whitney test. Statistically significant variables in the univariate analysis including median age and sex were entered into a multivariate model using logistic regression analysis, and the odds ratios (OR) and 95% confidence intervals (CI) were calculated. We selected median values in continuous variables to perform table 4. For instance, we selected 63 years because it was the median age of the cohort and the same reason was applied for the other variables. We clarified this on the methods section. 

Table 5: Why did you analyze 36 of transplanted patients? How did you exclude other 5 patients? What was the aim of this test as you have already given all statistical analysis with 41 transplanted patients?

R: In addition to usual statistical analysis to compare the transplant and non-transplant groups, we performed a propensity score match analysis to reduce possible bias between the two groups. For this reason, variables were balanced and we were only able to include 36 transplanted patients for the propensity score analysis. 

Discussion section:

-Why did not all transplanted patients have lopinavir/ritonavir treatment? Please give some details fort his condition.

R: Our first hospital protocol (from February to May 2020) consisted of lopinavir/ritonavir 400/100 mg BID for 7-14 days plus hydroxychloroquine 400 mg/12h on the first day, followed by 200 mg/12h for the next 4 days. Nevertheless, due to lopinavir/ritonavir interactions (especially with immunosuppressive therapy) some patients did not receive lopinavir/ritonavir as for their treating physician choice. We clarified that. Page 5, lines 120. 

-Hydroxychloroquine was advised to be not given at 04 July and lopinavir/ritonavir was advised to be not given at 06 July by World Health Organization. I respect your treatment at that time as most of medical practitioners used these drugs at that time. However, I think that you should add this fact to discussion section

R: We clarified this at the discussion section (page 20, line 307-309).

Reviewer #2: My concerns were below.

We would like to thank reviewer two for the very useful comments. We have incorporated the suggestions to our manuscript. 

Introduction

1. Authors should clearly define the purpose of the study and why these authors conducted this research.

R: According to your suggestion we included a final sentence in the introduction section. Page 3, lines 73-75. 

2. As I know that solid organ transplant recipients have more comorbid diseases than patients without SOT. The authors should point to this issue in a separate paragraph in this section.

R: We included a new paragraph. Page 3, lines 66-68.

Material and method

1. I recommend authors to use subtitles. Subheadings; patient selection, treatment, outcomes, hospitalization, and ICU admission may use.

R: Thank you for your suggestion; we have divided the methods section into paragraphs with subheadings, and incorporated subheadings in the tables. 

2. Inclusion and exclusion criteria should be clearly defined in this section.

R: Since it was an observational study, all patients admitted to our hospital with COVID-19 during the study period were included. We have clarified this in the methods section.

3. The selection of the patients without SOT should be defined in this section, and criteria for a propensity score match should be defined.

R: Propensity score matching was calculated using the following parameters: age, sex, hypertension, lung disease, use of anti-cytokine therapies, baseline OSD and OSD during hospitalization. We selected these variables because they were associated with worse outcomes in previous studies and we wanted to evaluate the effect of adjuvant therapies. 

4. Study endpoints should be clearly stated.

R: We have added the study endpoints at the introduction section. 

5. Authors should refer to the selection criteria of the patients without SOT.

R: All patients admitted to our hospital with COVID-19 during the study period were included. We have clarified this in the methods section

Results

1. The diabetes rate in Table 1 is misspelled.

R: Thank you for pointing it out, we have corrected the rate. 

2. After the propensity score match, the differences in covid-19 treatment should be explained.

R: Anti-cytokine therapy was one of the parameters used for the propensity score analysis, and therefore there were no differences between the groups. 

3. Kaplan Meier analysis can be useful for determining differences in mortality between groups.

R: According to your suggestion we performed a Kaplan Meier analysis. Figure 1.

Patients with macrophage activation syndrome due to COVID-19 can be shown in the tables.

R: Unfortunately we were unable to retrieve interleukin levels of all included patients. Instead we included other markers of a hyper inflammatory state such as ferritin levels, d-dimer and C-reactive protein. We believe these markers reflect common practices in most centers, as very few are able to quantify interleukins. 

 The hyperinflammatory syndrome observed in COVID-19 shares similarities with other hyperinflammatory disorders, such as secondary haemophagocytic lymphohistiocytosis, macrophage activation syndrome, macrophage activation-like syndrome of sepsis, and cytokine release syndrome. Nevertheless, these disorders, sometimes known as cytokine storm syndromes, share overlapping clinical manifestations and a common pathway of macrophage activation and a self-perpetuating cycle of cytokine production, but consensus agreement is lacking with regard to classification and diagnostic criteria. (Webb et al. Lancet Rheumatol. 2020 Dec; 2(12) ). 

Discussion

1. The mortality rate of SOT recipients with COVID-19 was lower than in some of the previous articles. The authors must find out what factors can be associated with these differences.

R: We have added the following paragraph to the discussion “Our mortality rate is similar to other transplant cohorts (3,4,10). However, other cohort have reported much worse outcomes, with mortality ranging from 28 to 67% (5,7). These differences might me explained by demographic differences, including limited heath resources in the context of an overstretched health system in the peak of the pandemic”. Page 21, lines 324-328

2. The authors should discuss the impact of chronic kidney disease on covid 19 mortality in a separate paragraph.

R: We have included baseline chronic kidney disease in both groups and added a sentence in the discussion. Page 21, line 340

3. Treatment modalities of the patients should be discussed in a separate paragraph.

R: We have included a new paragraph discussing COVID-19 therapies. Page 20, lines 301-314

4. The authors did not perform a power analysis before calculation, and the results may have been determined due to the second type of statistical error analysis. Therefore, authors should not use exact expressions.

R: We agree with reviewer and introduced limitations of this study at the discussion section, and have rephrased conclusions accordingly. 

Reviewer #3: This manuscript has conclusions similar to other large SOT cohorts that transplant itself does not necessarily increase risk for COVID-19 associated morbidity / mortality, and that other factors such as age and comorbidities play a role. While I agree with a propensity-matched score approach, the study has significant limitations including lack of information on key transplant-related variables (immunosuppression, timing from transplant, rejection) as well as the severity of illness in the non-transplant group.

R: We have included more variables related to transplantation, including immunosuppressive regimen, time from transplant and previous episodes of rejection (3 months prior to admission). Table 1.

Line 59: Mortality rate of 7% is higher than most estimates, although there are some studies were case fatality rate was that high. Suggest finding more accurate estimate.

R: We have included a paragraph in the discussion were we speculate on the rate of mortality of our cohort compared to other cohorts. Page 21, lines 324-328

Lines 89-97: Is there a reason that the authors did not use World Health Organization ordinal scale 0-8 that is current reference for clinical severity?

R: We agree with reviewer three that WHO ODS is the reference, and have changed it. 

Lines 103: What “anti-cytokine” therapy was used?

R: We have specified the anticytokine therapies available at our center at the time of our study in the methods section. 

Lines 131-132: how were the variables for propensity score matching chosen?

R: We performed the propensity score match to reduce possible bias between 2 groups and we chose age, gender, comorbidities, anti-cytokine therapy and mortality because we thought that these variables could be related with outcomes. In terms of treatment we only chose anti-cytokine therapy because no one in the transplant group received remdesivir. 

Line 140: What does “double transplant” mean? Is it a patient who has had 2 sequental transplants (e.g. 2 kidney transplants after a single failed grafts) or 2 organs (like kidney-pancreas)?

R: We have included in the results section, that 2 of the SOT patients were combined liver-kidney transplant. There were no sequential transplants. Page 7, lines 173.

Line 145: How was it determined that the patients had Strep pneumo co-infection? Was urinary antigen testing or culture used, and what imaging corresponded to make diagnosis?

R: To diagnose Strep pneumo co-infection the patients had to meet 2 criteria: 

1. Culture from the respiratory tract or positive urinary antigen.

2. Chest X-ray or CT scan suggestive of bacterial pneumonia. 

We added these diagnostic criteria to methods section. Page 5, lines 110-112. 

Line 146: Table 1 should include characteristics for the non-SOT group, with p-values. By definition, outcomes should not be included in Table 1. The authors are also clearly missing some important variables relevant to the degree of immunosuppression of SOT recipients such as time from transplant, induction regimen used, maintenance immunosuppression, presence / absence of rejection, recent treatment for rejection.

R: We included this information and deleted outcomes from table 1

Line 172: It is very interesting to compare the characteristics of SOT and non-SOT groups. It appears that the non-SOT group was quite sick with 23% of patients experiencing septic shock, and higher baseline mortality. This should be considered when interpreting the study

R: We have added a sentence in the discussion section pointing out this data. 

Line 260-263: The finding of differences in therapy is not significant other than the fact that it outlines the differences in practice at this one center.

R: We specifiied in the methods section the protocol that was being used in the time of the study, as recommendations on the treatment with COVID-19 continues to change. Physicians might be more cautious when using certain new antimicrobials in SOT patients, and drug interactions definitely play a role. We thought it was important to highlight these differences. 

Line 263-265: There ARE therapies that have been shown to be beneficial in RCTs: remdesivir and dexamethasone.

R: We changed this sentence, and have included references with the studies that support the use of these drugs. Page 20, lines 310-311. 

Line 279: The fact that transplant mortality was similar to other centers I think emphasizes that non-transplant mortality in this cohort was high.

R: The mortality rate of this cohort was studied during the first wave of COVID-19 was 14.5%. It does not differ from other series published in the same period. 

Line 298: Authors should include in limitation that there was routine use of agents without proven efficacy, and some with associated harm (protease inhibitors, hydroxychloroquine, anti-cytokine agents) were used as part of the institution’s standard of care.

R: We have added the following sentences: “We did find differences in the use of COVID-19 adjuvant therapy that was considered standard of care at the time of the study in the two groups. Transplant patients received significantly less lopinavir/ritonavir, and more anakinra compared to non-transplant patients. The important drug-drug interactions of lopinavir/ritonavir with immunosuppressant medications could have influenced physicians on prescribing this drug (12). Remdesivir which has been shown to reduce the time to clinical improvement, was not administered to SOT patients, possibly as a result of higher incidence, of chronic kidney disease and acute kidney injure, that contraindicate the use of this drug (13). Randomized control trials with lopinavir/ritonavir and azythromycin as adyuvant therapy for COVID-19 have shown no benefit, and some have shown deleterious effects, and therefore they are no longer being used”.

Comment: This manuscript appears to be written by non-native speaker. Please check for spelling and grammatical errors.

R: According to your suggestion the manuscript was revised by a native English speaker.

---

## [Decision Letter · Decision Letter 1]

25 Jan 2021

PONE-D-20-31759R1

No increased risk of mortality in solid organ transplant patients with COVID-19

PLOS ONE

Dear Dr. Bodro,

Thank you for submitting your manuscript to PLOS ONE. After careful consideration, we feel that it has merit but does not fully meet PLOS ONE’s publication criteria as it currently stands. Therefore, we invite you to submit a revised version of the manuscript that addresses the points raised during the review process.

We look forward to receiving your revised manuscript.

Kind regards,

Tzevat Tefik, MD

Academic Editor

PLOS ONE

Reviewers' comments:

Reviewer's Responses to Questions

**Comments to the Author**

1. If the authors have adequately addressed your comments raised in a previous round of review and you feel that this manuscript is now acceptable for publication, you may indicate that here to bypass the “Comments to the Author” section, enter your conflict of interest statement in the “Confidential to Editor” section, and submit your "Accept" recommendation.

Reviewer #1: All comments have been addressed

Reviewer #2: All comments have been addressed

2. Is the manuscript technically sound, and do the data support the conclusions?

Reviewer #1: Yes

Reviewer #2: No

3. Has the statistical analysis been performed appropriately and rigorously? 

Reviewer #1: Yes

Reviewer #2: Yes

4. Have the authors made all data underlying the findings in their manuscript fully available?

Reviewer #1: Yes

Reviewer #2: Yes

5. Is the manuscript presented in an intelligible fashion and written in standard English?

Reviewer #1: Yes

Reviewer #2: Yes

6. Review Comments to the Author

Reviewer #1: Dear authors,

Thank you very much for your efforts to revise the manuscript.

I think the final form of the article may be accepted for publication.

Reviewer #2: My concerns were below:

Title 

The title describes no increased risk of mortality, but authors must clearly define the group of the patient which, was shared the same risk of mortality between the solid organ recipient. I think that "No increased risk of mortality in solid organ transplant patients with COVID-19 compared to the non-solid organ transplant patients with COVID-19" or "A propensity score-matched analysis of mortality in solid organ transplant patients with COVID-19 compared to the non-solid organ transplant patients" may be better options.

Abstract

The authors defined the risk factors of mortality in solid organ transplant recipients. On the other hand, this issue is not written in this part. Authors should re-write or remove this sentence.

If this sentence is removed, criteria for a propensity score match would be useful for the readers.

 Introduction

Some of the major articles in this issue could not be referenced.

doi:10.1111/ajt.16185, 

doi:10.1056/NEJMc2011117, 

doi: 10.1097/TP.0000000000003533, 

doi:10.1097/TP.0000000000003433, 

doi:10.1111/ajt.16246, 

doi:10.1111/tid.13383, 

doi:10.1111/tid.13371, 

doi:10.1016/j.kint.2020.08.005 

maybe added to the references

Two sentences that began "The main objective" and "the aim of the study" were defined the same thing. The paragraph which began with "the main objective" may remove.

"Furthermore, besides immunosuppressive regimens, transplant recipients usually present more comorbidities such as hypertension and diabetes, possibly influencing the outcome of patients with COVID-19. On the other hand, any propensity score match analysis was not performed in mortality of solid organ recipients compared to non-solid organ transplant patients before." may be a better option.

Material and methods

Serum may be added before the laboratory measurements; such as C-reactive protein, creatinine.

All patients were followed-up at least after discharge for two months.

There must be an anticoagulation and ventilation protocol for COVID-19 which, must be described in the material and method section.

Result

The authors must describe why they measure in mean and SD for table 1 and median and IQR for table 2?

Twenty-one patients were diagnosed with pulmonary emboli, does the patient screen for pulmonary emboli, or are they have any symptoms? This issue must be described in the material and method section.

Table 2 should be re-calculated after propensity score-matched analysis.

Discussion

These mortality differences might be explained by the recipient and donor age, donor type, and the number of comorbid diseases.

One of the major limitations of the study, was the small sample size which, was not clearly described in this section. As I mentioned in the previous review, it may cause a type 2 error.

7. PLOS authors have the option to publish the peer review history of their article (what does this mean?). If published, this will include your full peer review and any attached files.

Reviewer #1: No

Reviewer #2: No

---

## [Author Response · Author response to Decision Letter 1]

25 Jan 2021

Dear editor, 

Thank you for considering our revised-manuscript entitled “No increased risk of mortality in solid organ transplant patients with COVID-19”. Following your instructions we have addressed the new comments from reviewer 2 in a point-to-point fashion that we are attaching. 

Review Comments to the Author

Reviewer #2: My concerns were below:

Thank you for your thorough review of our manuscript. We hope we have addressed all the concerns, and that the manuscript reads more clear now.

1) Title 

The title describes no increased risk of mortality, but authors must clearly define the group of the patient which, was shared the same risk of mortality between the solid organ recipient. I think that "No increased risk of mortality in solid organ transplant patients with COVID-19 compared to the non-solid organ transplant patients with COVID-19" or "A propensity score-matched analysis of mortality in solid organ transplant patients with COVID-19 compared to the non-solid organ transplant patients" may be better options.

Authors: thank you for the suggestion. We have modified the title accordingly to: "A propensity score-matched analysis of mortality in solid organ transplant patients with COVID-19 compared to the non-solid organ transplant patients".

2) Abstract

The authors defined the risk factors of mortality in solid organ transplant recipients. On the other hand, this issue is not written in this part. Authors should re-write or remove this sentence.

If this sentence is removed, criteria for a propensity score match would be useful for the readers.

Authors: We have included your suggestion in the abstract.

3) Introduction

Some of the major articles in this issue could not be referenced.

doi:10.1111/ajt.16185, 

doi:10.1056/NEJMc2011117, 

doi: 10.1097/TP.0000000000003533, 

doi:10.1097/TP.0000000000003433, 

doi:10.1111/ajt.16246, 

doi:10.1111/tid.13383, 

doi:10.1111/tid.13371, 

doi:10.1016/j.kint.2020.08.005 

maybe added to the references

Authors: We have added the suggested references. 

Two sentences that began "The main objective" and "the aim of the study" were defined the same thing. The paragraph which began with "the main objective" may remove.

Authors: We have deleted the paragraph that reviewer 2 mentioned. 

"Furthermore, besides immunosuppressive regimens, transplant recipients usually present more co-morbidities such as hypertension and diabetes, possibly influencing the outcome of patients with COVID-19. On the other hand, any propensity score match analysis was not performed in mortality of solid organ recipients compared to non-solid organ transplant patients before." may be a better option.

Authors: we have replaced the paragraph as suggested by reviewer 2

4) Material and methods

Serum may be added before the laboratory measurements; such as C-reactive protein, creatinine.

Authors: we have added serum before the laboratory measurements. 

All patients were followed-up at least after discharge for two months.

There must be an anticoagulation and ventilation protocol for COVID-19 which, must be described in the material and method section.

Authors: We have described the anticoagulation and ventilation protocol in the methods section. 

5) Results

The authors must describe why they measure in mean and SD for table 1 and median and IQR for table 2?

Authors: We have changed data. All measures were calculated in median and IQR.

Twenty-one patients were diagnosed with pulmonary emboli, does the patient screen for pulmonary emboli, or are they have any symptoms? This issue must be described in the material and method section.

Authors: We have specified that only certain patients are screened for PE, ie: if there is clinical suspicion. 

Table 2 should be re-calculated after propensity score-matched analysis.

Authors: In table 2 we compared the two groups before applying the propensity score-matched analysis. Comparison of transplant and non-transplant population after applying a propensity score analysis, using only variables we considered that could influence outcomes is shown in table 5. 

6) Discussion

These mortality differences might be explained by the recipient and donor age, donor type, and the number of co-morbid diseases.

One of the major limitations of the study, was the small sample size which, was not clearly described in this section. As I mentioned in the previous review, it may cause a type 2 error.

Authors: We have highlighted in the limitations the small sample size might cause a type 2 error.

---

## [Decision Letter · Decision Letter 2]

4 Feb 2021

A propensity score-matched analysis of mortality in solid organ transplant patients with COVID-19 compared to non-solid organ transplant patients

PONE-D-20-31759R2

Dear Dr. Bodro,

We’re pleased to inform you that your manuscript has been judged scientifically suitable for publication and will be formally accepted for publication once it meets all outstanding technical requirements.

Kind regards,

Tzevat Tefik, MD

Academic Editor

PLOS ONE

Additional Editor Comments (optional):

Reviewers' comments:

Reviewer's Responses to Questions

**Comments to the Author**

1. If the authors have adequately addressed your comments raised in a previous round of review and you feel that this manuscript is now acceptable for publication, you may indicate that here to bypass the “Comments to the Author” section, enter your conflict of interest statement in the “Confidential to Editor” section, and submit your "Accept" recommendation.

Reviewer #2: All comments have been addressed

2. Is the manuscript technically sound, and do the data support the conclusions?

Reviewer #2: Yes

3. Has the statistical analysis been performed appropriately and rigorously? 

Reviewer #2: Yes

4. Have the authors made all data underlying the findings in their manuscript fully available?

Reviewer #2: Yes

5. Is the manuscript presented in an intelligible fashion and written in standard English?

Reviewer #2: Yes

6. Review Comments to the Author

Reviewer #2: I believe that manuscript has been considerably improved, and grateful to the authors for their patience. I think that the article satisfied the standards for acceptance for PLOS ONE.

7. PLOS authors have the option to publish the peer review history of their article (what does this mean?). If published, this will include your full peer review and any attached files.

Reviewer #2: No

---

## [Editor Report · Acceptance letter]

18 Feb 2021

PONE-D-20-31759R2 

A propensity score-matched analysis of mortality in solid organ transplant patients with COVID-19 compared to non-solid organ transplant patients 

Dear Dr. Bodro:

I'm pleased to inform you that your manuscript has been deemed suitable for publication in PLOS ONE. Congratulations! Your manuscript is now with our production department. 

Kind regards, 

on behalf of

Dr. Tzevat Tefik 

Academic Editor

PLOS ONE